# Relation between Polygenic Risk Score, Vitamin D Status and BMI-for-Age *z* Score in Chinese Preschool Children

**DOI:** 10.3390/nu16060792

**Published:** 2024-03-11

**Authors:** Luolan Peng, Tingting Liu, Chao Han, Lili Shi, Chen Chen, Jinpeng Zhao, Jing Feng, Mengyao Wang, Qin Zhuo, Junsheng Huo, Yan Li, Zhaolong Gong

**Affiliations:** Key Laboratory of Trace Element Nutrition of National Health Commission, National Institute for Nutrition and Health, Chinese Center for Disease Control and Prevention, Beijing 100050, China; pengll0504@163.com (L.P.); liutt@ninh.chinacdc.cn (T.L.); hanchao@ninh.chinacdc.cn (C.H.); shill@ninh.chinacdc.cn (L.S.); chenchen@ninh.chinacdc.cn (C.C.); zhaojp@ninh.chinacdc.cn (J.Z.); fengjing0921@163.com (J.F.); romona123@163.com (M.W.); zhuoqin@ninh.chinacdc.cn (Q.Z.); huojs@ninh.chinacdc.cn (J.H.)

**Keywords:** polygenic risk score, vitamin D status, obesity, zBMI

## Abstract

Background: Both genetics and vitamin D deficiency are associated with childhood obesity. However, the role of vitamin D status between polygenic and childhood obesity has been unknown. The current study aimed to determine the relation between genetic factors, vitamin D status, and BMI-for-age *z* score (zBMI) in Chinese preschool children. Methods: A total of 1046 participants aged 3.7 to 6.6 years old from the Long-term Health Effects Assessment Project of Infants and Toddlers Nutritional Pack (LHEAPITNP) were included in this study. The polygenic risk score (PRS) was established based on 55 BMI-related single nucleotide polymorphisms (SNPs) derived from a published genome-wide association study (GWAS) for BMI. Serum 25(OH)D was used as an index of vitamin D status and measured with liquid chromatography-tandem mass spectrometry (LC/MS-MS) assay. The Wilcoxon test or Kruskal–Wallis test was used to compare the differences of variables between different groups and Spearman correlation analysis was used for analyzing the correlations between the PRS, 25(OH)D levels, and zBMI. Results: The PRS showed a positive relation to zBMI (*r*_s_ = 0.0953, *p* = 0.0022) and 25(OH)D showed a negative relation to zBMI (*r*_s_ = −0.1082, *p* = 0.0005) in the full-adjustment model. In addition, the differences in zBMI at different vitamin D statuses in the low-risk PRS group and the intermediate-risk PRS group were both statistically significant (*p*_low_ = 0.0308, *p*_intermediate_ = 0.0121), the median zBMI was both higher at vitamin D insufficiency status. And the difference in zBMI between different genetic risk groups was also statistically significant at vitamin D sufficiency status (*p* = 0.0077). Furthermore, genetic risk showed a positive relation to zBMI at vitamin D sufficiency status, and the *p* for trend was 0.0028. Conclusions: Our findings suggested that vitamin D was related to zBMI negatively in Chinese preschoolers and maintaining adequate vitamin D levels may only contribute to lower the zBMI in preschoolers with low and intermediate genetic susceptibility.

## 1. Introduction

The growing of childhood obesity has become a global public health problem. According to the report of the World Health Organization (WHO), an estimated 38.2 million children under the age of 5 years were overweight or obese in 2019 [1]. In China, the prevalence of overweight and obesity among children under 6 was 6.5% and 2.7% in 2002, while both increased to 6.8% and 3.6%, respectively according to the latest national prevalence estimates for 2015-19 [2]. Being overweight and obese not only affects the current physical and mental health of children but may also be a risk factor for adulthood obesity and chronic diseases such as fatty liver disease, cardiovascular disease, and type 2 diabetes [3,4].

Genetics play an important role in developing obesity. Around 40–70% of the variability in body mass index (BMI) has been attributed to genetic factors [5]. Based on genetic studies, obesity has usually been classified into two broad categories, monogenic obesity and polygenic obesity. Monogenic obesity is a rare and severe form of obesity that follows the Mendelian pattern of inheritance, whereas polygenic obesity, also known as common obesity, is influenced by a large number of polymorphisms, each having only a small effect [6]. Since the publication of the two genome-wide association studies (GWASs) on obesity in 2007 [7,8], which identified a cluster of single nucleotide polymorphisms (SNPs) associated with BMI in the first intron of the *FTO* locus, approximately 60 GWASs studies have been conducted and more than 1100 loci associated with a range of obesity traits have been identified [6]. Although most GWASs have been conducted in European populations, in recent years a growing number of GWASs have been performed in East Asian populations [9]. For example, a GWAS for BMI was performed on nearly 170,000 Japanese people and identified 51 novel loci associated with BMI [10]. Furthermore, most of the GWAS loci for obesity which were identified in adults first were also associated with obesity or BMI in children and adolescents [11,12,13].

However, most of these GWAS loci have a small effect and typically correspond to a small fraction of truly associated variants [14]. The polygenic risk score (PRS) method, which is also named genetic risk score (GRS), could aggregate the effects of variants across the genome and can be used to test for gene × environment and gene × gene interactions [15,16]. For example, Yoon et al. [17] developed a BMI-related PRS for predicting susceptibility to obesity and related traits in the Korean population. Additionally, obesity-related PRS/GRS have also been established to explore the associations between the genetic risk of obesity and environmental factors, such as lifestyle, dietary pattern, and neighborhood environment [18,19,20].

Vitamin D deficiency has also been increased globally and is associated with a series of adverse health outcomes, including obesity [21,22]. Some studies have explored vitamin D status concerning obesity and found an inverse relationship between vitamin D levels and BMI [23,24,25]. A recent meta-analysis suggested that vitamin D deficiency was associated with impaired lipid profiles among adults with overweight or obesity [26]. The study in children and adolescents showed similar results [27]. Genetic association analysis between 25-hydroxyvitamin D (25 [OH]D)-related genes and obesity has been conducted to explore genetic factors linked to vitamin D and obesity [28]. It has been reported that vitamin D receptor (VDR) gene polymorphisms were associated with childhood obesity and its adverse consequences [29].

However, research on the relationship between vitamin D status and the effect of obesity-related genes on obesity is limited. Additionally, most of the previous studies focused on European adults, with less research on children. And BMI for age is used to identify potentially wasted, overweight or obese children aged two years and older [30]. Our study therefore aimed to establish a BMI-related PRS and determine the relation between the BMI-related PRS, vitamin D status, and BMI-for-age *z* score (zBMI) in Chinese preschool children.

## 2. Materials and Methods

### 2.1. Study Participants

The Ministry of Health and the All-China Women’s Federation have jointly implemented the Nutrition Improvement Project on Children in Poor Areas of China (NIPCPAC) since 2012, providing a free pack of Ying Yang Bao (YYB), which contains a variety of essential nutrients and serves as home fortification for complementary feeding, for infants and young children (IYC) aged 6 to 24 months daily in poverty-stricken counties in China, to better the nutritional status of IYC [31]. Then, the Long-term Health Effects Assessment Project of Infants and Toddlers Nutritional Pack (LHEAPITNP), a prospective study, was launched in 2018 aiming to evaluate the short and long-term effects of YYB intervention in early life. It should be noted that children involved in this project have discontinued the intake of YYB.

In the four detailed monitoring locations of the LHEAPITNP, a total of 1104 children were invited to take part in the survey of 2022. The four detailed monitoring locations included Fuquan and Guiding in Guizhou Province, and Ruyang and Song County in Henan Province, China. As shown in Figure 1, 1046 participants aged 3.7 to 6.6 years old with valid data were included in the present study after quality control. The exclusion criteria included failure of DNA extraction, information missing, duplicate submissions, failure of genotyping, and outliers defined by outside mean ± 4 SD.

The LHEAPITNP was reviewed and approved by the Ethics Committee of the Institute of Nutrition and Health of the Chinese Center for Disease Control and Prevention (No. 2018-017). All the caregivers provided written informed consent.

### 2.2. BMI-for-Age z Score

After accurate measurement of participants’ height and weight with standardized equipment (the minimum scale for height measuring device was 0.1 cm and the minimum scale for electronic weight scales was 0.05 kg), BMI-for-age *z* score (zBMI) was calculated using WHO Anthro software (version 1.0.4) for children under 5 years and WHO Anthro Plus software (version 1.0.4) for children over 5 years, respectively.

### 2.3. Vitamin D Status

Serum 25(OH)D is the biomarker usually used as an index of vitamin D status [32]. In our study, serum 25(OH)D (D2+D3) was measured with liquid chromatography tandem mass spectrometry (LC/MS-MS) assay (LCMS-8060, SHIMADZU, Kyoto, Japan), which could measure 25(OH)D2 and 25(OH)D3, respectively. Serum samples obtained from centrifugation of blood samples were stored at −80 °C until measurement. Before LC-MS/MS analysis, all serum samples were pre-treated and mixed isotopic internal standard solutions and working solutions were prepared. The working solutions were diluted serially for preparation of calibration curves and quality control (QC) samples. To all serum samples, calibration samples and QC samples (200 μL, respectively), 20 μL internal standard solutions were added (100 ng/mL), and 400 μL methanol and acetonitrile mixture solutions (1:1) were also added to promote protein precipitation. Hexane (1.2 mL) was then added after 30 s of vortexing and shaking, followed by a further 5 min of vortexing and shaking. The resulting mixture was then centrifuged at 12,000× *g* for 5 min. Next, hexane was added, and the resulting mixture was centrifuged again. After the upper layer was transferred into the new tube, we dried it under a gentle stream of nitrogen at room temperature and re-dissolved it with 100 μL of mobile phase. It should be added that when the LC-MS/MS analysis was conducted the column temperature was 40 °C, the flow rate was set at 0.30 mL/min, and an Electrospray Ionization (ESI) source was used.

According to the Endocrine Society Clinician Vitamin D Guideline [33], vitamin D status was defined as insufficient (25(OH)D ≤ 30 ng/mL) and sufficient (25(OH)D > 30 ng/mL) in our analyses.

### 2.4. Genotyping and Single-Nucleotide Polymorphism (SNP) Selection

Genomic DNA was extracted using the magnetic beads method from blood clots, which were placed in the inert separator gel coagulation tubes and refrigerated at −80 °C after serum separation. Target single nucleotide polymorphisms (SNPs) were genotyped by Kompetitive Allele Specific PCR (KASP^TM^, LGC Genomics, Teddington, Middlesex, UK) [34]. Finally, genotyping data of all target SNPs were visualized with SNP Viewer software (version 2.0, Hoddesdon, UK).

A total of 85 SNPs that reached genome-wide significance (*p* < 5.0 × 10^−8^) were chosen from a GWAS of BMI in Japanese people (*n* = 173,430) [10]. Eight SNPs (rs10208649, rs183975233, rs148546399, rs4366055, rs10795945, rs12617004, rs11602339, rs7305242) were excluded from the score due to minor allele frequencies (MAFs) less than 0.01, thirteen SNPs (rs2076463, rs7020996, rs75766425, rs1379871, rs6529684, rs3121672, rs1190736, rs6433857, rs4357030, rs11030100, rs2540034, rs7903146, rs77511173) due to not being in Hardy–Weinberg Equilibrium (*p* < 1.0 × 10^−6^) and another nine SNPs (rs4308481, rs143665886, rs1832886, rs180950758, rs5945324, rs6913361, rs1846974, rs35560038, rs111612372) which were unavailable were also excluded. Finally, 55 SNPs were selected for the PRS. The sequences of the primers and detailed information for these 55 SNPs are available in the Appendix A.

### 2.5. Polygenic Risk Score

A total of 55 SNPs with an MAF ≥ 1% and missing rate ≤ 5% were selected to calculate BMI-related PRS. Every SNP was recorded as 0, 1, or 2 according to the number of effective alleles (alternative alleles) and weighted by its relative effect size (β-coefficient) obtained from the previously published GWAS [10]. The BMI-related PRS of each subject was calculated with the following equation: PRS = (β_1_ × SNP_1_ + β_2_ × SNP_2_ + … + β*_n_* × SNP*_n_*) × (*n*/sum of the β coefficients), where β is the effect size of each SNP, *n* is the number of each subject’s available SNPs, and sum of the β is the sum of the β coefficients of each subject’s available SNPs [18]. Further information of the effective allele of each SNP was available in Appendix A. Finally, the PRS was classified into low (bottom 10%)-risk, intermediate (10–90%)-risk, and high (top 10%)-risk groups in our analyses.

### 2.6. Covariates

Covariates were used for characteristic descriptions of participants and potential confounding adjustment of Spearman correlation analysis and were as follows: sex (Male/Female), age (months), birth length (cm), birth weight (g), premature birth (Yes/No), delivery (Unknown/Vaginal/Caesarean), only child (Yes/No), breastfeeding (Yes/No), vitamin D supplement (Yes/No), Ying Yang Bao (Yes/No), parental care (Yes/No), education of caregiver (Primary school or below/Junior middle school/High school or above) and socio-economic status (SES). Three points should be added: (1) “Vitamin D supplement” indicated whether participants had taken a vitamin D supplement in the week before the survey; (2) “Ying Yang Bao” indicated whether participants were from the YYB intervention group; and (3) SES was calculated based on the education and occupation of parents and was used to reflect economic status of participant’s family [35].

### 2.7. Statistical Analysis

Continuous variables were described using the median and 25th and 75th percentile in our study because of abnormal distribution and were compared by Wilcoxon test or Kruskal–Wallis test between different groups. To conduct multiple comparisons between different genetic risk groups, the *KC_WC* macro program (for SAS) was applied based on Dunn’s test for unequal sample sizes between different comparing groups [36]. Descriptive statistics of categorical variables were analyzed using the amount and related proportion (%), and comparisons among different groups were analyzed using the chi-square test. Spearman correlation analysis was used for analyzing the correlations between the PRS, 25(OH)D levels, and the zBMI. Model I was a crude model without adjustment for any covariates, model II was adjusted for sex, age, birth height, and birth weight, and model III was further adjusted for the remaining covariates.

To measure the associations between the PRS, 25(OH)D levels and the zBMI, we compared the median of zBMI at different vitamin statuses among different genetic risk groups. Similarly, we compared the median of zBMI at different genetic risk groups among different vitamin statuses. All analyses were performed by SAS version 9.4 (SAS Institute, Cary, NC, USA) and R (version 3.2), statistical significance was defined as a two-sided *p*-value less than 0.05.

## 3. Results

### 3.1. Demographic Characteristics of the Participants According to Genetic Risk

General characteristics of the 1046 participants were shown in Table 1. The median of zBMI was −0.45 among 528 boys and 518 girls aged 3.7 to 6.6 years old. The mean birth length was 50.1 cm, and the median birth weight was 3245 g. And of all children, 3.7% were born prematurely and 60.4% were born vaginally. A total of 13.3% were the only child in their family and 71.8% were raised primarily by their parents. In addition, more than half of caregivers had the highest level of junior middle school or above. And the median of SES was 47.8. Moreover, 89.6% of children were breastfed early in life. Regarding nutritional supplements, 49.8% of children were from the YYB intervention group and 3.4% had taken vitamin D supplements in the week before the survey. Finally, the median of the PRS and 25(OH)D levels were −19.25 and 33.63 ng/mL, respectively. After grouping participants according to their PRS, there were 104 individuals in the low-risk PRS group, 838 in the intermediate-risk PRS group, and 104 in the high-risk PRS group. As shown in Table 1, the PRS and zBMI were both statistically significantly different across genetic risk groups, while other variables were not.

### 3.2. Correlations between Polygenic Risk Score, 25(OH)D Levels, and zBMI

As shown in Table 2, in the full-adjustment model, the PRS established in our study showed a positive relation to zBMI (*r*_s_ = 0.0953, *p* = 0.0022). 25(OH)D showed a negative relation to zBMI (*r*_s_ = −0.1082, *p* = 0.0005) in the full-adjustment model. The correlations described above were in line with the results obtained in model I and model II.

### 3.3. Differences in zBMI at Different Genetic Risk Groups and Different Vitamin D Statuses

These results are shown in Table 3. Differences in zBMI at different genetic risk groups were statistically significant (*p* = 0.0029). There were statistical differences also found in the multiple comparisons. In comparison with the low-risk PRS group, the median zBMI was higher in the high-risk group. Additionally, statistical differences in zBMI were also found between the intermediate-risk group and the high-risk group. And difference in zBMI at different vitamin D statuses was also statistically significant (*p* = 0.0017). The median zBMI was higher at vitamin D insufficiency status.

### 3.4. Comparisons of zBMI across Different Subgroups

As shown in Table 4, the differences in zBMI at different vitamin D status in the low-risk PRS group and the intermediate-risk PRS group were both statistically significant (*p*_low_ = 0.0308, *p*_intermediate_ = 0.0121), while there was no statistical difference in zBMI at different vitamin D status found in the high-risk PRS group. Among all risk groups, the median zBMI was higher at vitamin D insufficiency status. In addition, the difference in zBMI between different genetic risk groups was statistically significant (*p* = 0.0077) at vitamin D sufficiency status. There were statistical differences also found in the multiple comparisons. Furthermore, at vitamin D sufficiency status, genetic risk showed a positive relation to zBMI, and the *p* for trend was 0.0028. Compared with the low-risk PRS group, the median zBMI was higher in the high-risk PRS group. And the median zBMI in the high-risk PRS group was also higher than it in the intermediate-risk PRS group. However, no interactive effects were found between genetic risk and vitamin D status on zBMI (*p* for interaction = 0.3912).

## 4. Discussion

We established a polygenic risk score consisting of 55 SNPs associated with BMI and it was related to the zBMI of preschoolers in our study. A negative relation between 25(OH)D levels and zBMI was also found. And when we explored the interplay between vitamin D status and the PRS, the findings suggested that maintaining adequate vitamin D levels may help lower zBMI for preschoolers with low and intermediate genetic susceptibility. Additionally, for children with adequate vitamin D levels, the PRS showed a significantly positive relation to zBMI, and a linear trend existed.

While we established a PRS to aggregate the effects of variants associated with BMI, to further explore the interactions between genetic risk and other environmental factors, our study was not the first to do this. Tyrrell et al. [18] established a BMI-related GRS (consisting of 69 SNPs) based on the genetic study of Locke et al. [5] and conducted gene–obesogenic environment interactions using more than 120,000 adults from the UK Biobank study. Then, they found that an obesogenic environment, especially relative social deprivation, could higher the risk of obesity for adults with genetic susceptibility. Mason et al. [37] also established the GRS (consisting of 91 SNPs) according to the study of Locke et al. [5] and found that adults with a higher genetic risk of obesity may be more vulnerable to fast-food accessibility. A similar study was also conducted on children: Fang et al. [38] selected 11 SNPs to establish the PRS from a study [12] which joined 20 GWASs, and the results suggested that adherence to a healthy lifestyle during childhood may lower the genetic susceptibility to obesity. However, the PRS established by Fang et al. was derived from individuals of European ancestry but was used to examine the genetic risk of obesity for Chinese children, which ignored the complexities of trans-ancestry. Cumulative evidence has indicated that PRS models trained with European individuals were less accurate when applied to other ethnic populations compared to the European populations [39]. Moreover, some studies have demonstrated that the genetics of obesity are relatively constant over the course of a lifetime [11,12,13]. Therefore, we established our PRS based on individuals of East Asian ancestry [10] and applied it to Chinese preschoolers. And the PRS showed a positive relation to zBMI of children in our study.

Additionally, an inverse relationship between 25(OH)D levels and zBMI was shown in our findings, and it was consistent with previous studies. Saneei et al. [23] conducted a meta-analysis involving 34 cross-sectional studies and found a significant inverse weak correlation between 25(OH)D levels and BMI in adults. A meta-analysis including a total of 55 observational studies demonstrated that vitamin D levels were negatively associated with BMI both in diabetic and non-diabetic subjects, while a more significant correlation was seen in the diabetic subject population [25]. Furthermore, several studies suggested that increasing vitamin D levels could help improve the lipid profile of adolescents or children, and vitamin D deficiency may impair the lipid profile of adults [26,27]. While multiple studies have found a negative correlation between vitamin D levels and obesity, the direction of this relationship has remained a subject of controversy. [40]. A bi-directional Mendelian randomization analysis of multiple cohorts [24] was conducted and suggested that increased BMI may lead to low 25(OH)D, but lower 25(OH)D levels had minimal effect on BMI increase. Mallard et al. [41] conducted a meta-analysis including randomized and nonrandomized controlled trials and found a significant weak association between weight loss and higher 25(OH)D. However, no significant changes in 25(OH)D of subjects were found after bariatric surgery [42,43]. Equally, there is still no convincing evidence to support the effect of vitamin D supplementation on body weight. Unfortunately, our results only showed an inverse relationship between 25(OH)D levels and zBMI in preschool children and did not provide further evidence on the influence of vitamin D status on obesity or whether obesity leads to vitamin D deficiency, so more intervention trials are needed in the future to answer this question.

To further explore the molecular mechanism of vitamin D in obesity, genetic studies of the association between vitamin D deficiency and obesity have been conducted in recent years, focusing mainly on the relationship between vitamin D-related genes and obesity [28]. A genetic association analysis [44] investigated the contribution of the vitamin D receptor (*VDR*) genetic variants (*TaqI*, *BsmI* and *FokI*) to several obesity-related traits and found that *VDR* genetic variants were not significantly associated with obesity-related phenotypes in Caucasian young adults. And Wang et al. [45] also studied the association of *VDR* gene with metabolic syndrome and found that *VDR* gene polymorphisms may be correlated with obesity or metabolic syndrome in Chinese children. Furthermore, the related molecular mechanism of the relationship between *VDR* gene polymorphisms and obesity is available in the review of Akter et al. [29].

However, few studies have focused on whether obesity-related genes influence vitamin D levels and the interaction between obesity-related polymorphisms and vitamin D status on childhood obesity. Therefore, we also conducted correlation analysis on PRS and 25(OH)D levels and explored the interplay between genetic risk and vitamin D status on the zBMI of children. Eventually, no significant relation between the PRS and 25(OH)D levels was found in our study. And the results showed that within each group of genetic risk, zBMI was lower in participants with sufficient vitamin D. But significant difference in zBMI between different vitamin D statuses was only seen in the low-risk PRS group and the intermediate-risk PRS group, which indicated that maintaining sufficient vitamin D may only help lower the zBMI in preschoolers with low and intermediate genetic risk, and not in those with high genetic risk.

To our knowledge, this is the first study to determine the relationship between the BMI-related PRS, vitamin D levels, and zBMI in Chinese preschoolers. We not only aggregated the effects of many variants for the following analyses but also avoided the trans-ancestry problem. Furthermore, both Han and non-Han children were included in the study, so that the study covered a wider population. However, there are not without limitations to our study. First, no causal inferences could be drawn from this study as it was based on cross-sectional data. Second, our study was limited to preschoolers aged 3.7 to 6.6 years from China rural areas, and the generalizability of our findings should be tested in other demographic populations. Third, some unavailable variables that may influence results were not included in the analysis, such as the BMI of the mother and sun exposure of subjects. Moreover, *p* < 5.0 × 10^−8^ was chosen as the selection threshold directly, the PRS were therefore not calculated over a range of thresholds, which may result in poor PRS with high standard error. Finally, the explaining variation of zBMI by the PRS reached only 1.0%, which was calculated based on the linear regression by the ranks of zBMI and the PRS among current participants. In the study of Locke et al. [5], the 97 SNPs associated with BMI explained 2.7% of the variance in BMI, which compared to a relatively small percentage of variance in zBMI explained by our established PRS.

## 5. Conclusions

Our results provide new evidence for association studies between genetics, vitamin D levels, and zBMI in preschoolers. An inverse relationship between 25(OH)D levels and zBMI was found in Chinese preschoolers. Our results indicated that maintaining adequate vitamin D levels may only contribute to lower the zBMI in preschoolers with low and intermediate genetic susceptibility and not in those with high genetic risk. Further studies therefore should be performed to further determine the interaction between vitamin D status and genetic risk of childhood obesity and provide effective prevention and treatment strategies.

## Figures and Tables

**Figure 1 nutrients-16-00792-f001:**
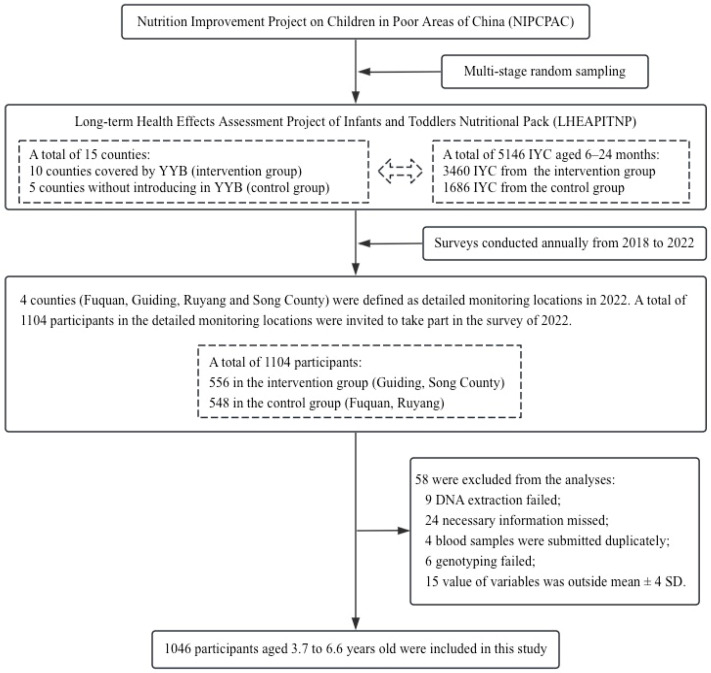
Process of including eligible participants in the present study.

**Table 1 nutrients-16-00792-t001:** Comparisons of the low-, intermediate-, and high-risk PRS groups for the characteristics.

Variables *	Total (*n* = 1046)	Genetic Risk Groups		
		Low-Risk PRS (*n* = 104)	Intermediate-Risk PRS (*n* = 838)	High-Risk PRS (*n* = 104)
Sex, *n* (%)				
Male	528 (50.5)	56 (53.8)	423 (50.5)	49 (47.1)
Female	518 (49.5)	48 (46.2)	415 (49.5)	55 (52.9)
Age (months)	61.6 (57.3, 66.2)	61.9 (57.0, 67.2)	61.6 (57.4, 66.2)	60.7 (56.3, 66.3)
Birth length (cm) ^†^	50.1 ± 1.4	50.4 ± 1.4	50.1 ± 1.4	50.0 ± 1.5
Birth weight (g)	3245 (3000, 3500)	3245 (3000, 3600)	3245 (3000, 3500)	3222 (3000, 3500)
Premature birth, *n* (%)				
Yes	39 (3.7)	6 (5.8)	28 (3.3)	5 (4.8)
No	1007 (96.3)	98 (94.2)	810 (96.7)	99 (95.2)
Delivery, *n* (%)				
Unknown	11 (1.1)	1 (1.0)	9 (1.1)	1 (1.0)
Vaginal	632 (60.4)	56 (53.8)	513 (61.2)	63 (60.6)
Caesarean	403 (38.5)	47 (45.2)	316 (37.7)	40 (38.4)
Only child, *n* (%)				
Unknown	33 (3.1)	4 (3.9)	28 (3.3)	1 (1.0)
Yes	139 (13.3)	10 (9.6)	108 (12.9)	21 (20.2)
No	874 (83.6)	90 (86.5)	702 (83.8)	82 (78.8)
Breast feeding, *n* (%)				
Yes	937 (89.6)	93 (89.4)	753 (89.9)	91 (87.5)
No	109 (10.4)	11 (10.6)	85 (10.1)	13 (12.5)
Vitamin D supplement, *n* (%)				
Yes	36 (3.4)	5 (4.8)	26 (3.1)	5 (4.8)
No	1010 (96.6)	99 (95.2)	812 (96.9)	99 (95.2)
Ying Yang Bao, *n* (%)				
Yes	521 (49.8)	47 (45.2)	423 (50.5)	51 (49.0)
No	525 (50.2)	57 (54.8)	415 (49.5)	53 (51.0)
Parental care, *n* (%)				
Yes	751 (71.8)	77 (74.0)	602 (71.8)	72 (69.2)
No	295 (28.2)	27 (26.0)	236 (28.2)	32 (30.8)
Education of caregiver, *n* (%)				
Primary school or below	306 (29.3)	30 (28.9)	245 (29.2)	31 (29.8)
Junior middle school	519 (49.6)	54 (51.9)	415 (49.5)	50 (48.1)
High school or above	221 (21.1)	20 (19.2)	178 (21.3)	23 (22.1)
SES	47.8 (44.8, 50.8)	47.8 (44.8, 50.8)	47.8 (44.8, 50.9)	47.8 (44.8, 51.4)
PRS **	−19.25 (−55.79, 14.54)	−110.20 (−123.95, −96.25)	−19.25 (−48.32, 7.46)	69.34 (58.73, 83.68)
25(OH)D levels (ng/mL)	33.63 (26.79, 41.46)	35.58 (26.87, 44.93)	33.73 (26.92, 41.24)	31.5 (25.16, 40.33)
zBMI **	−0.45 (−1.04, 0.15)	−0.53 (−1.15, 0.10)	−0.47 (−1.04, 0.11)	−0.11 (−0.87, 0.73)

Genetic risk groups were classified according to the PRS; the bottom 10% was defined as low-risk PRS, the top 10% was defined as high-risk PRS, and the remaining was defined as intermediate-risk PRS. Abbreviations: SES, socioeconomic status; PRS, polygenic risk score; 25(OH)D, 25-hydroxyvitamin D; zBMI, BMI-for-age *z* score. * Values of categorical variables may not sum to 100% due to rounding. ** Indicates *p*-value < 0.05. Continuous variables were described as P50 (P25, P75) due to abnormal distribution and categorical variables were described as amounts with proportions. ^†^ Since the P25 and P75 of birth length were almost identical in each group, the birth length was finally described with mean and standard error, and the P50 (P25, P75) of each group were added: total 50.0 (50.0, 50.0), the low-risk PRS group 50.0 (50.0, 50.8), the intermediate-risk PRS group 50.0 (50.0, 50.0), the high-risk PRS group 50.0 (50.0, 50.0).

**Table 2 nutrients-16-00792-t002:** Correlations between the polygenic risk score, 25(OH)D levels, and zBMI.

Correlation	Model I ^a^		Model II ^b^		Model III ^c^	
	*r* _s_	*p*-Value	*r* _s_	*p*-Value	*r* _s_	*p*-Value
PRS and 25(OH)D levels	−0.0307	0.3210	−0.0340	0.2731	−0.0367	0.2388
PRS and zBMI	0.0912	0.0031	0.0961	0.0019	0.0953	0.0022
25(OH)D levels and zBMI	−0.1084	0.0004	−0.1051	0.0007	−0.1082	0.0005

Abbreviations: PRS, polygenic risk score; 25(OH)D, 25-hydroxyvitamin D; zBMI, BMI-for-age *z* score. ^a^ Crude model without adjusting any covariables; ^b^ Adjusted for age, sex (Male/Female), birth length (cm), and birth weight (g); ^c^ Further adjusted for premature birth (Yes/No), delivery (Unknown/Vaginal/Caesarean), only child (Yes/No), breastfeeding (Yes/No), vitamin D supplement (Yes/No), Ying Yang Bao (Yes/No), parental care (Yes/No), education of caregiver (Primary school or below/Junior middle school/High school or above) and socioeconomic status (SES).

**Table 3 nutrients-16-00792-t003:** Difference in zBMI at different genetic risk groups and different vitamin D statuses.

Group	*N*	zBMI	*p*-Value
Polygenic risk score			0.0029
Low-risk PRS ^a^	104	−0.53 (−1.15, 0.10)	
Intermediate-risk PRS ^b^	838	−0.47 (−1.04, 0.11)	
High-risk PRS	104	−0.11 (−0.87, 0.73)	
Vitamin D status			0.0017
Vitamin D insufficiency	393	−0.36 (−0.95, 0.32)	
Vitamin D sufficiency	653	−0.52 (−1.09, 0.04)	

Abbreviations: PRS, polygenic risk score; zBMI, BMI-for-age *z* score. “25(OH)D ≤ 30 ng/mL” was defined as vitamin D insufficiency and “25(OH)D > 30 ng/mL” was defined as vitamin D sufficiency. ^a^ Difference in zBMI between the low-risk PRS group and the high-risk PRS group was statistically significant; ^b^ Difference in zBMI between the intermediate-risk PRS group and the high-risk PRS group was statistically significant.

**Table 4 nutrients-16-00792-t004:** Differences in zBMI at different vitamin D status across different genetic risk groups and differences in zBMI between different genetic risk groups at different vitamin D statuses.

Subgroup	*N*	zBMI	*p*-Value
Vitamin D status by PRS			
Low-risk PRS			0.0308
Vitamin D insufficiency	36	−0.34 (−0.9, 0.45)	
Vitamin D sufficiency	68	−0.75 (−1.23, −0.08)	
Intermediate-risk PRS			0.0121
Vitamin D insufficiency	315	−0.38 (−0.99, 0.29)	
Vitamin D sufficiency	523	−0.54 (−1.09, 0.02)	
High-risk PRS			0.6216
Vitamin D insufficiency	42	0.04 (−0.81, 0.76)	
Vitamin D sufficiency	62	−0.2 (−0.88, 0.48)	
PRS by vitamin D status			
Vitamin D insufficiency			0.1874
Low-risk PRS	36	−0.34 (−0.9, 0.45)	
Intermediate-risk PRS	315	−0.38 (−0.99, 0.29)	
High-risk PRS	42	−0.04 (−0.81, 0.76)	
Vitamin D sufficiency			0.0077
Low-risk PRS ^a^	68	−0.75 (−1.23, −0.08)	
Intermediate-risk PRS ^b^	523	−0.54 (−1.09, 0.02)	
High-risk PRS	62	−0.2 (−0.88, 0.48)	

Abbreviations: PRS, polygenic risk score; zBMI, BMI-for-age *z* score. “25(OH)D ≤ 30 ng/mL” was defined as vitamin D insufficiency and “25(OH)D > 30 ng/mL” was defined as vitamin D sufficiency. ^a^ Difference in zBMI between the low-risk PRS group and the high-risk PRS group was statistically significant; ^b^ Difference in zBMI between the intermediate-risk PRS group and the high-risk PRS group was statistically significant.

## Data Availability

The datasets generated and analyzed during the current study are not publicly available but are available from the corresponding author on reasonable request.

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
