# Peer review of "Relation between Polygenic Risk Score, Vitamin D Status and BMI-for-Age z Score in Chinese Preschool Children"

_nutrients, 2024, doi:10.3390/nu16060792_

Round 1

Reviewer 1 Report (Previous Reviewer 1)

Comments and Suggestions for Authors

The authors have addressed all the concerns raised by me during the first review session. Moreover, the study population has been extended. However, one thing is weird, specifically the inclusion of non-Han children, specified as an exclusion criterion in the first version of the manuscript. The inclusion of this ethnic group should be mentioned as a strength or a limitation of the study.

Author Response

Reviewer 2 Report (Previous Reviewer 2)

Comments and Suggestions for Authors

Thank you for your try to revision.

Even though there are still limitation to explain to some part, it is worthy to try.

Author Response

This manuscript is a resubmission of an earlier submission. The following is a list of the peer review reports and author responses from that submission.

Round 1

Reviewer 1 Report

Comments and Suggestions for Authors

The manuscript entitled "Relation between polygenic risk score, vitamin D status and BMI-for-age z score in Chinese preschool children" reports the results of an association study between vitamin D status and the genetic background of children.

The study has been performed on a high number of participants, thus it has statistical power, however, some methodological concerns could be raised.

First, the LC-MS/MS method used for 25(OH)D2 and 25(OH)D3 quantification should be described in more detail, because it is widely known that the lack of validity of the analytical method for 25(OH)D quantification can undermine the findings. Please specify the sample preparation technique, chromatographic conditions (column, retention time, temperatures), and MS conditions (ESI or APCI, m/z), unless you have used a previously validated and published method.

Second, although the value of PRS in disease risk prediction is becoming increasingly evident, the development and validation of a PRS should be performed before it can be applied. Since the authors of the current study arbitrarily selected 57 SNPs to calculate the PRS for BMI, some details should be provided regarding the shrinkage method of effect size. According to Choi et. al, 2020, the use of unadjusted effect size estimates of all SNPs could generate poorly estimated PRSs with high standard error. Please comment on the development and validation (?) PRS on target data. Eventually, specify the limitations of the PRS.

Third, not all the conclusions are supported by the data presented. Some parts should be rephrased.

Lines 109-110 "standardized equipment" - it should be specified.

References should be updated. Yoon et al 2023 10.3390/ijms241411560 and PMC6021129 should be included. 

  Comments on the Quality of English Language

Minor editing required.

Reviewer 2 Report

Comments and Suggestions for Authors

Thank you for your contribution to our journal.

This study is to make an effort to evaluate the relation between PRS, vitamin D and zBMI in Chinese preschool children. This is very interesting study, however, I want to several question for this study. 

1. Is there any maternal effect? eg, BMI of mother, vitamin D intake during pregnance, etc.

2. What about sun-exposure of the subjects?

3. If you explain the reason of significant result in only low-risk PRS group, it would be more understandable.

4. In this cross-sectional finding, especially 25OHD, it is not easy to explain of this all results, even though no vitamin D supplementation in this subjects. As you commented, if your data have VDR polymorphism, it would show better relation.
